# Test-Time Self-Adaptive Small Language Models for Question Answering

**Soyeong Jeong**[1]    **Jinheon Baek**[2]    **Sukmin Cho**[1]    **Sung Ju Hwang**[1,2]    **Jong C. Park**[1*]

School of Computing[1]    Graduate School of AI[2]
Korea Advanced Institute of Science and Technology[1,2]
{starsuzi,jinheon.baek,nelllpic,sjhwang82,jongpark}@kaist.ac.kr

## Abstract

Recent instruction-finetuned large language models (LMs) have achieved notable performances in various tasks, such as question-answering (QA). However, despite their ability to memorize a vast amount of general knowledge across diverse tasks, they might be suboptimal on specific tasks due to their limited capacity to transfer and adapt knowledge to target tasks. Moreover, further finetuning LMs with labeled datasets is often infeasible due to their absence, but it is also questionable if we can transfer smaller LMs having limited knowledge only with unlabeled test data. In this work, we show and investigate the capabilities of smaller self-adaptive LMs, only with unlabeled test data. In particular, we first stochastically generate multiple answers, and then ensemble them while filtering out low-quality samples to mitigate noise from inaccurate labels. Our proposed self-adaption strategy demonstrates significant performance improvements on benchmark QA datasets with higher robustness across diverse prompts, enabling LMs to stay stable. Code is available at: https://github.com/starsuzi/T-SAS.

## 1 Introduction

Language models (LMs) have gained the ability to learn generalizable representations that are applicable to diverse tasks by being trained on massive text corpora with increased parameters (Brown et al., 2020; Kojima et al., 2022). Moreover, to enhance the transferability to unseen tasks, LMs are further fine-tuned on instructions that are verbalized from a vast amount of the supervised datasets, showing a remarkable zero-shot ability across a wide range of tasks (Wei et al., 2022a; Sanh et al., 2022).

However, despite their ability to store a vast amount of general knowledge across diverse tasks, LMs show suboptimal performances on specific downstream tasks when transferring and adapting

---

*Corresponding author

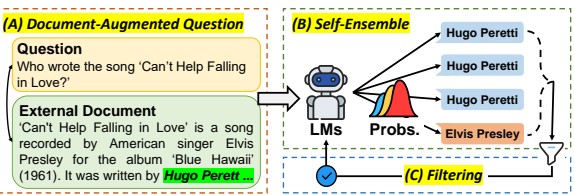

Figure 1: Illustration of our proposed T-SAS that includes self-ensemble and filtering strategies for self-adapting LMs.

their knowledge to target tasks. One possible solution is to additionally fine-tune LMs, but this is often impractical in realistic scenarios where labeled datasets are scarce. Furthermore, while large LMs with hundreds of billions of parameters may solve specific target tasks without fine-tuning, they are rarely accessible. Motivated by these challenges, we focus on investigating the self-adaptive capabilities of *smaller LMs* during the test-time.

While several studies have shed light on the potential for self-improvement of LMs, they mainly focus on augmenting the supervised labeled data with additional self-generated labels (He et al., 2020; Chen et al., 2023; Wang et al., 2023b), which significantly differs from ours with a more challenging setup that does not rely on labeled data. Note that there exists a recent work (Huang et al., 2022) that has shown the self-adaptive ability of large LMs with 540B parameters by further training with their generated answers, focusing on the reasoning tasks. However, the utilization of large LMs requires substantial costs and restricted accessibility. Therefore, our attention shifts towards smaller LMs, whose potential to be adapted to target tasks with their limited capability of storing knowledge remains largely unexplored.

Note that the adaptation of smaller LMs to downstream tasks brings forth new and critical challenges. First, it seems evident that large LMs with extensive knowledge are adept at facilitating self-adaption. However, the question arises regarding the self-adaptive capability of smaller LMs, specifically when the supervised labeled datasets are absent. Second, it may be suboptimal to use all the

self-generated labels for training, as some of them could contain possibly incorrect information, leading to significant performance degradation (Zhou et al., 2023). Third, unlike the large LMs, adapting smaller LMs to specific tasks would require the incorporation of external knowledge, due to their limited capabilities of storing specific knowledge.

In this work, we aim at addressing the challenges of self-adaptive abilities in smaller LMs without additional labeled datasets, focusing on the QA tasks augmented with external knowledge. To this end, we first stochastically generate multiple answers according to a given unlabeled question with its related document. Then the most plausible answer is selected through majority voting, serving as a pseudo-label for training during test-time. Note, however, that we should be aware of the possibility of unreliable results from the self-ensemble. To mitigate this, we further propose to filter out potentially incorrect samples based on low agreement among self-generated labels, as illustrated in Figure 1. We refer to our proposed method as **T**est-time **S**elf-**A**daptive **S**mall LMs (T-SAS). We validate our T-SAS on QA datasets with smaller LMs, on which T-SAS significantly improves the self-adaptive capabilities of smaller LMs.

Our contributions and findings are threefold:

- We first explore the self-adaptive capability of the smaller LMs in a realistic setting only with the unlabeled data during the test-time.
- We ensure that the high quality of the self-generated labels is maintained by proposing a novel self-ensemble scheme with filtering.
- We show that our T-SAS method achieves outstanding performance on the QA tasks.

## 2 Related Work

**Language Models**   Pre-trained language models have shown decent advancements in diverse tasks (Brown et al., 2020) and further improved by increasing the number of parameters to billions (Touvron et al., 2023; Anil et al., 2023) and leveraging instruction-finetuning techniques with a vast amount of supervised labeled datasets (Wei et al., 2022a; Sanh et al., 2022). Despite their recent successes (Wei et al., 2022b; Wang et al., 2023a), however, we see that LMs still encounter difficulties in effectively addressing downstream tasks due to their limited task-specific adaptation.

**Self-adaptive LMs**   Due to the frequent occurrence of domain or data shift in real-world scenar-

ios, self-adaptive models have gained substantial attention (Wang et al., 2021a; Shu et al., 2022; Veksler, 2023). In particular, some work has suggested to augment supervised labeled training data with self-generated labels (He et al., 2020; Chen et al., 2023; Wang et al., 2023b). In contrast, we assume a more realistic test-time setup without labeled data. Note that there are recent studies on self-consistent LMs. Specifically, Huang et al. (2022) demonstrated that large LMs can be further trained with the most consistent labels among multiple self-generated labels. On the other hand, our focus lies on more practical and accessible smaller LMs with novel strategies of answer sampling and filtering. While some research focuses on self-consistent prompts by regularizing LMs to generate consistent outputs across different prompts (Zhou et al., 2022; Zeng and Gao, 2023; Wan et al., 2023), the focus is different and orthogonal to ours which proposes to self-adapt smaller LMs for specific target tasks associated with external knowledge.

**Self-adaption for Extractive QAs**   Several previous studies explored the self-adaptive capabilities of the traditional pre-trained language models, but they mainly addressed classification problems under the extractive setting (Li et al., 2020; Shakeri et al., 2020; Banerjee et al., 2021; Wang et al., 2021b; Ye et al., 2022). However, we focus on the generative setting, which makes a large difference due to fundamentally different objectives. To be specific, in an extractive setting, self-adaptation is based on probabilities, while in a generative setting, self-adaptation is done using generated text. In situations where filtering is further applied, filtering is based on probabilities in the extractive setting, whereas, it is done using the generated text in the generative setting. Furthermore, Li et al. (2020) and Shakeri et al. (2020) assume an unsupervised QA setting, where the context-question-answer triplets are not available, thus requiring an additional query-generation module. Such a pair generation approach is different and orthogonal to ours, since we aim to enhance answer generation directly from the provided context and question.

## 3 Method

### 3.1 Preliminaries

**Question Answering**   Let $\mathcal{D}_{train}$ be a labeled QA training set, where each instance consists of a question $q_i$, a gold answer $a_i^*$, and its associated doc-

Table 1: Exact Match (EM) and F1 scores on three QA benchmark datasets with varying sizes of FLAN.

| Datasets | Methods | Base (250M) | | Large (780M) | | XL (3B) | |
|---|---|---|---|---|---|---|---|
| | | EM | F1 | EM | F1 | EM | F1 |
| **NQ** | Finetuned w/ Training Set | 67.47 | 75.28 | 73.41 | 80.79 | 74.36 | 81.72 |
| | Naïve LM w/o Ext. | 3.50 | 7.20 | 7.19 | 12.24 | 12.09 | 18.24 |
| | Naïve LM | 37.21 | 45.00 | 53.76 | 65.34 | 54.53 | 66.39 |
| | Self-Adaptive w/ Greedy | 32.75 | 40.69 | 56.78 | 67.69 | 59.34 | 71.04 |
| | Self-Adaptive w/ Soft | 39.53 | 49.24 | 54.29 | 65.98 | 58.17 | 71.03 |
| | Self-Adaptive w/ LMSI | 39.81 | 49.36 | 56.79 | 68.08 | 62.87 | 74.17 |
| | T-SAS (Ours) | **41.70** | **50.22** | **63.90** | **74.20** | **63.96** | **75.29** |
| **TQA** | Finetuned w/ Training Set | 71.78 | 77.28 | 80.84 | 85.01 | 81.93 | 86.40 |
| | Naïve LM w/o Ext. | 5.96 | 11.54 | 13.29 | 19.57 | 25.30 | 31.14 |
| | Naïve LM | 51.83 | 60.79 | 69.77 | 76.69 | 75.27 | 81.17 |
| | Self-Adaptive w/ Greedy | 52.95 | 60.98 | 69.05 | 75.45 | 76.94 | 82.72 |
| | Self-Adaptive w/ Soft | 49.57 | 58.31 | 69.02 | 76.19 | 75.15 | 81.64 |
| | Self-Adaptive w/ LMSI | 56.55 | 65.45 | 70.47 | 77.42 | 77.34 | 83.37 |
| | T-SAS (Ours) | **60.67** | **67.93** | **74.46** | **80.60** | **78.38** | **84.01** |
| **SQD** | Finetuned w/ Training Set | 69.94 | 81.72 | 74.26 | 85.56 | 75.58 | 86.55 |
| | Naïve LM w/o Ext. | 2.00 | 6.37 | 3.55 | 9.01 | 5.86 | 12.17 |
| | Naïve LM | 59.93 | 71.25 | 68.33 | 80.22 | 71.66 | 83.00 |
| | Self-Adaptive w/ Greedy | 57.20 | 68.93 | 64.62 | 76.77 | 73.03 | 84.03 |
| | Self-Adaptive w/ Soft | 51.79 | 65.48 | 65.78 | 79.02 | 71.13 | 83.41 |
| | Self-Adaptive w/ LMSI | 60.42 | 72.57 | 69.19 | 80.87 | 73.42 | 84.39 |
| | T-SAS (Ours) | **63.02** | **74.68** | **71.34** | **82.57** | **73.84** | **84.72** |

ument $d_i$ that contains $a_i^*$, as follows: $\mathcal{D}_{train} = \{(q_i, d_i, a_i^*)\}$ with $a_i^* \in d_i$. Similarly, an unlabeled test set is defined as follows: $\mathcal{D}_{test} = \{(q_i, d_i)\}$. Assume that LM is a language model parameterized with $\theta$, which has been instruction-finetuned on massive datasets. Then, given a pair of the question and its relevant document $(q_i, d_i)$, LM generates an answer, as follows: $\bar{a}_i = \mathsf{LM}(d_i, q_i; \theta)$. Note that, in order to generate a correct answer, i.e., $a_i^* = \bar{a}_i$, it is beneficial to train LM with a labeled training set by minimizing the loss (e.g., cross-entropy) between a correct answer $a_i^*$ and model prediction $\bar{a}_i$ as follows: $\mathcal{L}(a_i^*, \bar{a}_i)$. Then, after a training phase, LM is more likely to correctly predict $a_i^*$ on the unlabeled test set $\mathcal{D}_{test}$ of the target task.

**Self-Adaptive LM**   While recent LM is capable of answering questions, directly using LM to the target task may yield suboptimal results, thus requiring transfer learning or adaption. In order to do so, our idea is to maximally leverage the unlabeled test set $\mathcal{D}_{test}$ that we have in hand for the target task. In particular, we have $\mathrm{D}_{test}$, and given that, a possible solution is to train LM with its self-generated pseudo label, $\bar{a}_i^*$. In other words, LM can be further trained on $\mathcal{D}_{test\_self} = \{(q_i, d_i, \bar{a}_i^*)\}$, where $\bar{a}_i^*$ is generated from the unlabeled test sample $(q_i, d_i)$ with LM, for self-adaption to target tasks.

## 3.2 Test-time Self-Adaptive Small LMs

We describe our test-time self-adaptive LMs (T-SAS) with proposed strategies for effectively generating and utilizing pseudo labels during test-time.

**Stochastic Self-Ensemble**   Note that relying on a single $\bar{a}_i^*$ generated by LM, which is trained on general domains, may result in inaccurate predictions when adapted to the target task, as there exists a possibility of incorrectly self-generated $\bar{a}_i^*$. To mitigate this, we propose to make LM generate multiple answers $\{\bar{a}_{i,j}\}_{j=1}^n$ with diverse points of view. Note that, while existing work (Huang et al., 2022; Wang et al., 2023a) proposed to use Top-$k$ or nucleus sampling (Fan et al., 2018; Holtzman et al., 2020) when generating $\{\bar{a}_{i,j}\}_{j=1}^n$ for reasoning tasks, their diversities might be limited due to answer sampling based on a single representation. Instead, we propose to leverage *multiple representations* generated through Monte-Carlo (MC) dropout (Gal and Ghahramani, 2016) by randomly masking weights on LM during test-time, as follows:

$$\{\bar{a}_{i,j}\}_{j=1}^n = \{\mathsf{LM}_{M \sim \mathcal{M}}(d_i, q_i; \theta \odot M)\}_{j=1}^n, \quad (1)$$

where $\mathcal{M}$ is a distribution of mask weights and $M$ is a sampled mask weight. Once we have generated multiple answers $\{\bar{a}_{i,j}\}_{j=1}^n$, our next objective is to assign one pseudo label $\bar{a}_i^*$ from the set using a majority voting strategy, which selects $\bar{a}_i^*$ with the highest number of occurrences among $\{\bar{a}_{i,j}\}_{j=1}^n$. After acquiring the self-generated label $\bar{a}_i^*$ for the unlabeled $\mathcal{D}_{test}$, we now aim at training LM on the $\mathcal{D}_{test\_self}$ with a following loss term: $\mathcal{L}(\bar{a}_i^*, \bar{a}_i)$.

**Filtering**   However, in contrast to Huang et al. (2022) and Wang et al. (2023a) who use all samples and their associated pseudo labels $\bar{a}_i^*$ for training, relying entirely on them can be largely prob-

| Datasets | Methods | EM | F1 |
|----------|---------|-----|-----|
| SciQ | Naïve LM | 71.80 | 80.31 |
| | T-SAS (Ours) | **73.40** | **81.30** |
| cpgQA | Naïve LM | 51.69 | 72.93 |
| | T-SAS (Ours) | **53.42** | **74.30** |
| TyDiQA | Naïve LM | 64.55 | 76.97 |
| | T-SAS (Ours) | **67.95** | **79.31** |

Table 2: Results on additional QA datasets with FLAN-T5-XL: SciQ, cpgQA, and TyDiQA (only using the English data).

| Methods | Large (780M) | XL (3B) |
|---------|--------------|---------|
| T-SAS (Ours) | **74.20** | **75.29** |
| w/o Stochastic | 67.67 | 74.10 |
| w/o Filtering | 68.25 | 73.24 |
| Naïve LM | 65.34 | 66.39 |

Table 3: Ablation studies on the NQ dataset with FLAN-T5-Large and FLAN-T5-XL based models .

lematic since LM lacks specific training to make valuable predictions on particular tasks. Therefore, motivated by the fact that the substantial performance improvements of LMs are primarily attributed to their training on high-quality data (Zhou et al., 2023), we further propose an automatic filtering strategy to identify and exclude samples, $\{(q_i, d_i, \bar{a}_i^*)\} \subset \mathcal{D}_{test\_self}$, that are likely to have incorrect $\bar{a}_i^*$. We determine this by removing samples labeled with $\bar{a}_i^*$ that have a vote count proportionally lower than a certain threshold.

# 4 Experiments

## 4.1 Experimental Setups

In this subsection, we describe experimental setups. Further details are shown in **Appendix** A.

**Datasets** We use three QA datasets, augmented with external documents from Wikipedia, pre-processed by Karpukhin et al. (2020): **1) Natural Questions (NQ)** (Kwiatkowski et al., 2019), **2) TriviaQA (TQA)** (Joshi et al., 2017), and **3) SQuAD v1.1 (SQD)** (Rajpurkar et al., 2016).

**Baselines and Our Model** We compare our T-SAS against other baselines using unlabeled test data. We use the FLAN (Chung et al., 2022) and the same prompt across all models. **1) Finetuned w/ Training Set** is an indicator model, which is finetuned on the labeled training set. **2) Naïve LM w/o Ext.** is a naïve baseline without external knowledge and self-adaptive training. **3) Naïve LM** is a baseline without self-adaptive training, but incorporates external knowledge. **4) Self-Adaptive w/ Greedy** is trained on the self-generated pseudo-labels via Greedy decoding. **5) Self-Adaptive w/ Soft** is trained on the result of soft voting. **6) Self-Adaptive w/ LMSI** is trained on the result of majority voting, using Top-$k$ sampling (Huang et al., 2022). **7) T-SAS (Ours)** is ours, which incorporates both self-ensembling and filtering strategies.

## 4.2 Results

Here, we show the overall performance of T-SAS. Please see **Appendix** B for more results.

**Main Results** As Table 1 shows, T-SAS significantly outperforms all baselines of varying sizes, particularly with a FLAN-Large model. Note that Naïve LMs show substantially lower performance than the supervision finetuned LMs. This corroborates our hypothesis that LMs are not fully optimized for target tasks. Also, by integrating external knowledge, the performance of all models, especially smaller ones, is largely improved.

Furthermore, ensembling multiple self-generated predictions significantly enhances performance, compared to baselines with Greedy decoding. This indicates that T-SAS, considering diverse points of view, reduces the likelihood of encountering performance-degrading scenarios caused by relying on a single prediction.

However, when compared to the baselines with multiple self-generated predictions (i.e., Self-Adaptive w/ Soft and w/ LMSI), the results indicate that a filtering strategy is required. Solely relying on the final result of the self-ensemble should be avoided with smaller LMs, contrasting the observation by Huang et al. (2022) with large LMs (540B) on the reasoning task. Interestingly, the model trained on soft labels, which must leverage all the predictions, shows even lower performance than the Naïve LMs, emphasizing the negative impact of training on inaccurate self-generated labels.

Comparing the performance with large LMs is outside our scope, since our work targets at investigating the effectiveness of smaller LMs on self-adaption. Note that large LMs and smaller LMs are not directly comparable to each other, due to their discrepancy in capacity. However, we additionally report the performance of the large zero-shot LM as an indicator in Figure 2. Surprisingly, our T-SAS significantly outperforms a much larger zero-shot LM, which further signifies the effectiveness of the proposed self-adaptive strategies for smaller LMs.

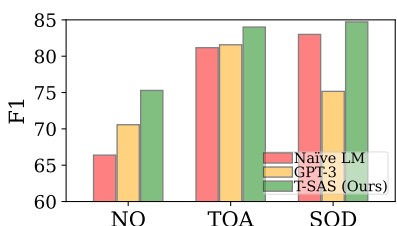
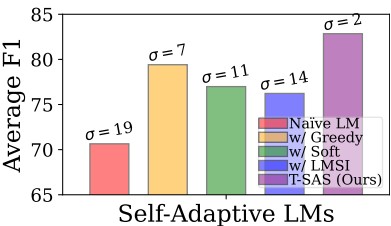
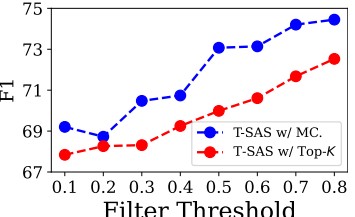

Figure 2: Results on three QA datasets with FLAN-T5-XL based models and GPT-3 (text-davinci-003).

Figure 3: Robustness to the diverse prompts, on the TQA using the FLAN-T5-XL models with 11 different prompts.

Figure 4: Comparison of MC drop and Top-$k$ sampling with varying thresholds on NQ with FLAN-T5-Large.

**Robustness on Diverse Prompts**   Recent LMs have been observed to be prompt-sensitive (Cho et al., 2023; Ishibashi et al., 2023), which is undesirable as consistent outputs are expected across similar prompts, particularly in real-world scenarios with diverse users. As shown in Figure 3, T-SAS shows substantial robustness to diverse prompts, which are attributed to the proposed stochastic generation and filtering strategies. In other words, T-SAS effectively mitigates the impact of inaccurate predictions by filtering them from multiple perspectives, even for specific prompts.

**Effectiveness on Specific Domains**   In addition to evaluating T-SAS in general domains, we further conduct experiments on specific domains, including the science domain, SciQ (Welbl et al., 2017), and the clinical domain, cpgQA (Mahbub et al., 2023), to assess its domain adaptability. Table 2 shows consistent improvements from T-SAS, highlighting its ability to capture and enhance transferability across domain shifts. These findings indicate the applicability of T-SAS in domains where high-quality labeled data is scarce.

**Effectiveness on Unseen Datasets**   Recent instruction-finetuned language models have been extensively trained on various QA datasets, making it challenging to evaluate them on new datasets. Therefore, in Table 2, we further show clear improvements even on the unseen datasets, cpgQA and TyDiQA (Clark et al., 2020). Also, it is worth noting that the performance improvement achieved by applying self-adaptation to the already trained data is not our weakness, but rather a strength. To be more specific, even though an LM has been trained comprehensively on diverse datasets including the target dataset, the LM remains as the general-purpose model and is not tailored to the specific target dataset. However, by using our proposed T-SAS on the target dataset, the model can further achieve improved performance thanks to its self-adaptation capabilities over the target dataset.

**Stochastic Generation Strategy with Filtering**
We compare MC dropout and Top-$k$ sampling with varying filtering thresholds. As shown in Figure 4, both strategies benefit from a filtering strategy by mitigating noises introduced during stochastic generation processes. Moreover, MC dropout consistently outperforms Top-$k$ sampling, which can be attributed to its higher lexical diversity (0.24 vs. 0.14). These findings suggest that diversity, combined with a filtering strategy, allows LMs to effectively identify and remove low-quality outputs with higher variances, consequently resulting in overall performance improvement.

**Ablation Studies**   In order to see how each of the stochastic generation and filtering strategies contributes to the overall performance, we provide the ablation studies with two variants of T-SAS with Large and XL sizes. To be specific, in T-SAS w/o Stochastic, low-quality samples are filtered out based on the generation probability of a single prediction, and in T-SAS w/o Filtering, the majority voting results are directly used as pseudo-labels without applying our proposed filtering strategy. As shown in Table 3, both of the proposed strategies positively contribute to the overall performance, by substantially improving the performance of Naïve LMs in both model sizes. Furthermore, these strategies indicate that they are in a complementary relationship, suggesting that they work together to enhance overall performance.

## 5   Conclusion

In this work, we investigated and improved the self-adaptive capabilities of the recent smaller LMs only using unlabeled test data, on the QA task. Specifically, our proposed method involves self-ensembling the stochastically generated labels and a filtering strategy to remove possibly incorrect labels, thereby enabling training with high-quality self-generated labels. The experimental results and analyses indicate that our method significantly improves QA performances during test-time.

## Limitations

While we show clear advantages of using our T-SAS to address the realistic challenges of the recent LMs regarding their self-adaptive capabilities, it is important to acknowledge that our validation was conducted under the assumption of having gold external documents containing the answers. However, in real-world scenarios, obtaining such gold documents may not be feasible, necessitating the incorporation of additional retrieval modules to retrieve query-relevant information. Although the integration of retrieval modules is beyond the scope of our current work, our exploration of the potential benefits of an external knowledge-augmented setting for self-adaptive smaller LMs opens up fruitful avenues for future research. We also believe that investigating the integration of retrieval modules with T-SAS to further enhance the practical applicability of self-adaptive LMs in real-world applications holds a significant value.

## Ethics Statement

The experimental results confirm the effectiveness of T-SAS in adapting to unlabeled test-time data. However, it is important to consider and address the potential bias of the underlying LM when utilizing its extensive knowledge for training self-adaptive models. We suggest that implementing specific filtering strategies targeted at mitigating these biases can be a potential solution to ensure the safety and reliability of self-adaptive models.

## Acknowledgements

This work was supported by Institute for Information and communications Technology Promotion (IITP) grant funded by the Korea government (No. 2018-0-00582, Prediction and augmentation of the credibility distribution via linguistic analysis and automated evidence document collection) and Basic Science Research Program through the National Research Foundation of Korea (NRF) funded by the Ministry of Education (RS-2023-00275747).

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

| Sizes | Methods | EM | F1 |
|---|---|---|---|
| **Small (80M)** | **Naïve LM** | 23.98 | 31.92 |
| | **T-SAS (Ours)** | **31.28** | **40.05** |
| **XXL (11B)** | **Naïve LM** | 60.73 | 70.93 |
| | **T-SAS (Ours)** | **66.52** | **76.58** |

Table 4: Results of FLAN-Small and FLAN-XXL, on NQ.

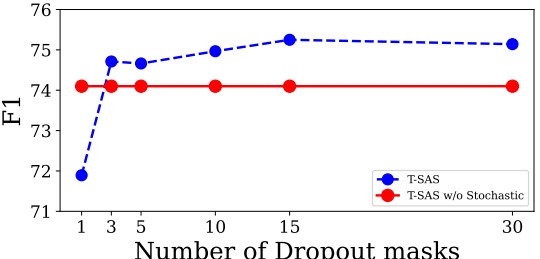

Figure 5: F1 scores with varying numbers of dropout masks.

## A Experimental Setups

**Datasets**   We use three QA datasets, augmented with external documents from Wikipedia, preprocessed by Karpukhin et al. (2020). **1) Natural Questions (NQ)** (Kwiatkowski et al., 2019) is composed of questions from the Google Search engine. **2) TriviaQA (TQA)** (Joshi et al., 2017) is collected from the Web, which consists of trivia questions. **3) SQuAD v1.1 (SQD)** (Rajpurkar et al., 2016) is constructed by the annotators by writing questions after reading passages.

**Metrics**   We evaluate models with Exact Match (EM) and F1-score (F1), following the standard protocol for the QA task (Rajpurkar et al., 2016; Karpukhin et al., 2020).

**Implementation Details**   For a fair comparison, we compare the models using the instruction-finetuned FLAN-T5 (Chung et al., 2022) model with three different sizes, Base (250M), Large (780M), and XL (3B) with the same prompt: *'Read this and answer the question{context}{question}'*. For the models larger than 3B, we trained them adopting a low-rank adaptation (LoRA) method (Hu et al., 2022), For hyperparameters, we set the training epoch as 5 for all the self-adaptive models and 1 for the indicator model. Also, we set the number of stochastically generated predictions as 15 and set the filtering threshold as 0.7.

## B Additional Experimental Results

**Dropout Mask Variations**   To investigate the impact of the number of masks in MC dropout on performance, we conduct experiments with varying mask numbers. Figure 5 illustrates that increasing the number of masks leads to improved performance, but stabilized after reaching a certain number of masks. Note that the stochastically perturbed models, with multiple dropout masks, offer more diverse perspectives compared to the ablated model without stochastic generation or the model with only one dropout mask, thus resulting

in decreasing the possibility of selecting inaccurate self-generated answers.

**Effectiveness on Diverse Model Sizes**   In addition to the performance on three sizes shown in Table 1, we additionally conduct experiments for the FLAN-Small (80M) and FLAN-XXL (11B) models. The results in Table 4 demonstrate that our T-SAS consistently enhances performances on the much smaller or larger model sizes.

**Effectiveness on T0-3B**   In addition to the FLAN series, we conduct experiments on another LM, T0 (Sanh et al., 2022) with 3B size. As shown in Table 5, our T-SAS consistently improves the performance on three QA datasets, which indicates the applicability of T-SAS on diverse LMs.

**Effectiveness of Augmented External Document**
We have observed significant performance improvement with the augmented external documents in all models as shown in Table 1. Here, we further analyze the importance of augmenting external documents, especially for the models that require training. As shown in Table 6, the overall performance without external knowledge is largely devastated for all models. Interestingly, the performance degradation for an indicator model trained on the supervised training dataset is remarkable. These findings corroborate our proposed challenge that external knowledge is necessarily required for training self-adaptive and especially smaller LMs, whose capabilities of storing specific knowledge are highly limited.

**Case Study**   We conduct a case study, mainly comparing our T-SAS against a self-adaptive baseline model with LMSI, in Table 7. The first example shows the robustness of our T-SAS approach in addressing prompt-sensitive situations, where an LM exhibits significantly degraded performance for a specific prompt, *'Read the following article and answer the question. Article: {} Question: {}'*. While both models stochastically generate in-

| Datasets | Methods | EM | F1 |
|---|---|---|---|
| NQ | Naïve LM | 42.08 | 54.78 |
| | T-SAS (Ours) | **50.17** | **61.61** |
| TQA | Naïve LM | 62.43 | 70.23 |
| | T-SAS (Ours) | **68.70** | **75.49** |
| SQuAD | Naïve LM | 50.15 | 62.73 |
| | T-SAS (Ours) | **56.86** | **69.10** |

Table 5: Results with T0-3B on three QA datasets.

| Methods | EM | F1 |
|---|---|---|
| **Naïve LM** | 54.53 | 66.39 |
| **Naïve LM w/o Ext.** | 12.09 | 18.24 |
| **Finetuned. w/o Ext.** | 7.16 | 12.79 |
| **T-SAS (Ours) w/o Ext.** | 12.29 | 18.43 |

Table 6: Results without using external documents, on NQ.

accurate predictions that are totally unrelated to the question or document, and subsequently vote based on these erroneous labels, our T-SAS prevents further training with these unreliable predictions by employing a filtering strategy. This demonstrates the adaptability of our T-SAS in real-world scenarios with diverse user prompts. Moreover, the second example highlights the effectiveness of a filtering strategy in addressing diverse answers generated by our stochastic self-generation strategy. Note that MC dropout produces answers with higher lexical diversity than Top-$k$ sampling, which enables the removal of low-quality outputs with a higher variance. In both cases, the removal of low-quality labels is crucial for the self-adaptive LMs, and our proposed strategies have demonstrated the effectiveness of achieving this goal.

Table 7: Examples from two self-adaptive LMs without using labeled data, during the test-time.

---

**Case # 1:** Our T-SAS exhibits robustness in handling diverse prompts.

**Document**: · · · This was necessary because EMI owned another record label called Columbia, which operated in every market except North America, Spain and Japan. CBS sold the record company in 1988 to **(Answer) Sony**. In 1991, the CBS label was officially renamed Columbia Records and the company was renamed Sony Music Entertainment. · · ·
**Question**: Which Japanese company bought CBS records in 1988?
**Prompt**: Read the following article and answer the question. Article: {context} Question: {question}

| Self-Adaptive w/ LMSI | T-SAS (Ours) |
|---|---|
| **Self-Generated Answers**: 'iv.', '(C).', 'a).', '(4).', '(a).', 'iv.', '[ii]', '[d].', '[iv]', 'sony', '(iv).', '(iii).', 'B).', 'iv.', 'sony' | **Self-Generated Answers**: '[iv]', 'b).', '(4).', '(iv).', '[i]', 'd).', '[i]', '[ii]', 'sony', 'sony', '[i]', '[iv]', '[ii]', '[ii]', '(iv)' |
| **Self-Ensemble Result:** 'iv.' | **Self-Ensemble Result:** ~~'[i]'~~ ($\frac{3}{15} <$ Threshold) |
| **Final Prediction:** '[iv]' | **Final Prediction:** 'sony' |

---

**Case # 2:** Our T-SAS shows effectively combines diverse answer generation and filtering strategies.

**Document**: · · · Howard in the 2007 film Spider-Man 3 and by **(Answer) Emma Stone** in the 2012 reboot film The Amazing Spider-Man and its sequel The Amazing Spider-Man 2. Created by writer Stan Lee and artist Steve Ditko, Gwen Stacy first appeared in The Amazing Spider-Man # 31 (December 1965). In her initial appearances, Peter Parker met Gwen while both were studying as undergraduates at Empire State University, but with Aunt May in the hospital, Peter was troubled and ignored her advances. · · ·
**Question**: Real name of Gwen Stacy in Amazing Spiderman
**Prompt**: Read this and answer the question {context} {question}

| Self-Adaptive w/ LMSI | T-SAS (Ours) |
|---|---|
| **Self-Generated Answers**: 'peter parker's intellect', 'peter parker's scientific rigor and', 'peter parker's interest in science', 'peter parker', 'peter', 'peter parker's scientific approach to solving mysteries', 'peter parker's intelligence', 'peter parker', 'peter parker', 'peter parker', 'peter's intellect and', 'peter parker', 'peter parker's scientific prowess', 'peter parker's intellect and sass', 'peter parker's understanding of science' | **Self-Generated Answers**: 'howard', 'gwen stacy', 'howard', 'howard', 'emma stone', 'howard', 'gwen stacy', 'emma stone', 'peter parker's wit and intelligence', 'howard', 'howard', 'gwen stacy', 'emma stone', 'emma stone', 'emma stone' |
| **Self-Ensemble Result:** 'peter parker' | **Self-Ensemble Result:** ~~'howard'~~ ($\frac{6}{15} <$ Threshold) |
| **Final Prediction:** 'peter parker' | **Final Prediction:** 'emma stone' |