# OpenReview forum: "Test-Time Self-Adaptive Small Language Models for Question Answering"
_EMNLP/2023/Conference — EMNLP 2023 Findings_

### Official Review · Reviewer_kW16 · 2023-07-31

**Soundness:** 3

**Excitement:**

3: Ambivalent: It has merits (e.g., it reports state-of-the-art results, the idea is nice), but there are key weaknesses (e.g., it describes incremental work), and it can significantly benefit from another round of revision. However, I won't object to accepting it if my co-reviewers champion it.

**Missing References:**

https://aclanthology.org/2020.acl-main.600.pdf

https://aclanthology.org/2020.emnlp-main.439.pdf

**Paper Topic And Main Contributions:**

This paper proposes a “self-adaptive” technique which enables smaller LM fine-tuning without labeled data. The authors introduce a novel sampling technique which uses Monte-Carlo dropout to create many potential answers for an unlabelled question which allows for improved filtering of noisy answers. They provide thorough empirical evidence showing that their proposed sampling and filtration method is more effective than previous methods.

**Reasons To Accept:**

- Exploring the ability of open-source models to perform on QA tasks without supervision is an important and impactful research direction.
- Proposed sampling methodology is interesting and novel.
- Empirical results are sound, baselines are strong and thorough and all the necessary ablations are present.

**Reasons To Reject:**

- The first reason for rejection I see is that at least ⅔ datasets used in this study are part of the training set used to instruction-tune Flan-T5 models. The authors do not discuss the implications of this fact in the paper, which essentially could change the self-adaptive or self-training setting into more of a semi-supervised or even data-augmentation setting. This could be grounds for rejection.
- The second important reason for rejection would be that the current work is not contextualized thoroughly enough. There is some mention of recent self-training works such as Wang et al. 2023 or Chen et al. 2023 but it is unclear how this work provides a “more realistic test-time setup without labeled data”. This paper is very related to QA self-training papers like the following but no discussion about this is added.
   - https://aclanthology.org/2020.acl-main.600.pdf
   - https://aclanthology.org/2020.emnlp-main.439.pdf
- To be clear, I believe this paper does expand on the current literature but it still requires more explicit contextualization and more clarity in its limitations.
- Minor issues:
   - FLAN and FLAN-T5 are different models, please clearly state that you are using FLAN-T5 throughout the paper.
   - The tables and figures on page 4 are very crammed, you should have just one caption with (left, center, right) divisions or split the figures up.

**Reproducibility:**

2: Would be hard pressed to reproduce the results. The contribution depends on data that are simply not available outside the author's institution or consortium; not enough details are provided.

**Reviewer Confidence:**

3: Pretty sure, but there's a chance I missed something. Although I have a good feel for this area in general, I did not carefully check the paper's details, e.g., the math, experimental design, or novelty.

---

> ### Author Rebuttal · Authors · 2023-08-29
>
> >**Weakness 1-1)** The first reason for rejection I see is that at least ⅔ datasets used in this study are part of the training set used to instruction-tuned Flan-T5 models.
>
> Please note that FLAN-T5 series models have gained recognition for their extensive training across nearly all available benchmark QA datasets, thus making it difficult to evaluate them with previously unseen benchmark QA datasets. However, it is important to mention that these models were not previously trained on certain QA datasets (e.g., cpgQA in Table 2), and we experimentally show that our T-SAS exhibits clear improvements even on them.
>
> Additionally, we have extended our analysis to include experiments on another unseen QA dataset, namely TyDiQA [C3], only using the English data in Table C.1. As shown in Table C.1, the proposed T-SAS model further demonstrates substantial enhancements in performance.
>
> Also, it is worth noting that the performance improvement achieved by applying self-adaptation to the already trained data is not our weakness, but rather a strength. To be more specific, even though an LM has been trained comprehensively on diverse datasets including the target dataset, the LM remains as the general-purpose model and is not tailored to the specific target dataset. However, by using our proposed T-SAS on the target dataset, the model can further achieve improved performance thanks to its self-adaptation capabilities to the target dataset.
>
> Table C.1: Results on the unseen dataset, TyDiQA with English, with FLAN-T5-XL based models.
> |Methods|EM|F1|
> |---|---|---|
> | Naïve LM | 64.55 | 76.97 |
> | T-SAS | **67.95** | **79.31** |
>
> [C1] TyDi QA: A Benchmark for Information-Seeking Question Answering in Typologically Diverse Languages
>
> >**Weakness 1-2)** The authors do not discuss the implications of this fact (i.e., at least ⅔ datasets used in this study are part of the training set used for instruction fine-tuning Flan-T5) in the paper, which could be grounds for rejection.
>
> Please refer to **Weakness 1-1** for a detailed answer. We will discuss these implications in the revision. Specifically, in Section 4.2, we will include the experimental results of the additional unseen data (Table C.1). Moreover, in the Limitation section, we will further discuss the challenge associated with evaluating FLAN-T5 series models on unseen benchmark QA datasets due to their limited availability. We hope that these updates, requiring minor revisions, will address your concerns.
>
>
> >**Weakness 2)** More discussion on the other related work [C2-C3].
>
> We thank you for suggesting related work. However, please note that the suggested papers and ours are not directly comparable for the following reasons:
>
> * **Extractive QA vs Generative QA**:  The suggested papers target at the extractive QA setting, while our T-SAS is designed under a generative QA setting. Since the extractive QA models aim at extracting exact answers from the given context, they address the problem as the token classification task. Please note that as the objectives of extractive and generative settings are different, the test-time adaption strategies are fundamentally different. To be specific, in an extractive setting, adaptation is based on probabilities, while in a generative setting, adaptation is done using generated text. In situations where filtering is further applied, in the extractive setting, filtering is based on probabilities, whereas in the generative setting, it is done using the generated text (We discovered that it is less effective to filter out samples based on a single generation probability in Table 3). This makes it difficult to directly compare our generative T-SAS against the extractive models. Therefore, while several previous work has investigated self-training approaches to solving classification tasks under the extractive setting, it has been questionable and largely underexplored how we can self-adapt the small LMs over the generative setting.
>
> * **Question-Answer pair generation vs Answer generation**: As the suggested papers assume an unsupervised QA setting, the context-question-answer triplets are not available. Therefore, they are required to first generate question-answer pairs with an additional query-generation module. Also, such a pair generation approach is different and rather orthogonal to ours, since we aim to enhance answer generation directly from the provided context and question.
>
>
> We concretely further discuss the differences between each of the suggested papers and ours as follows:
>
> [C2] Harvesting and Refining Question-Answer Pairs for Unsupervised QA
> * This work is an **Extractive** task (QA), which is largely different from our generative setting, as aforementioned above.
> * This work focuses on the **Question-Answer pair generation** strategy, requiring an additional query generation module.
> * In addition, the filtering approach in this work is based on a single token probability. Please note that such a filtering approach applied to our generative setting is already presented as a baseline in Table 3 (w/o Stochastic), which indicates the effectiveness of our proposed stochastic filtering strategy.
>
> [C3] End-to-End Synthetic Data Generation for Domain Adaptation of Question Answering Systems
> * This work is an **Extractive** task (QA), which is largely different from our generative setting, as aforementioned above.
> * This work focuses on the **Question-Answer pair generation** strategy, requiring an additional query generation step.
> * Furthermore, we already compared the filtering approach used in this work as a baseline in Table 3 (w/o Stochastic), which is based on the likelihood of the generated question-answer pair, showing that our proposed filtering strategy outperforms this baseline.
>
>
> >**Weakness 3)** The paper requires more explicit contextualization and more clarity in its limitations.
>
> Please refer to **Weakness 2** for more explicit contextualization and **Weakness 1** for more clarity on limitations. To summarize the answers on them, regarding contextualization, we will further discuss how our proposed T-SAS is different and orthogonal to the suggested unsupervised learning scheme and question-answer pair generation work in Section 2. Also, we will further clarify that the setting of test-time adaption under the generative QA setting is largely underexplored. Furthermore, we will discuss the problems of the limited availability of the unseen QA benchmark datasets in the Limitation section, as mentioned in **Weakness 2-1**. We believe that these revisions are minor to address and hope that you are satisfied with them.
>
>
> >**Weakness 4)** Minor issues of terminology and presentation.
>
> Thank you for your suggestion and we apologize for the confusion. We will update them in the revision.

---

### Official Review · Reviewer_FMJN · 2023-08-01

**Soundness:** 4

**Excitement:**

3: Ambivalent: It has merits (e.g., it reports state-of-the-art results, the idea is nice), but there are key weaknesses (e.g., it describes incremental work), and it can significantly benefit from another round of revision. However, I won't object to accepting it if my co-reviewers champion it.

**Paper Topic And Main Contributions:**

Although large language models show a strong ability to store a vast amount of general knowledge across diverse tasks, they struggle with suboptimal performances on specific downstream tasks when transferring and adapting their knowledge to target tasks. Fine-tuning LMs is a possible solution, but it is impractical when labeled datasets are scarce.

This work investigates the capabilities of smaller self-adaptive LMs, only with unlabeled test data. The author devises a self-adaption strategy that stochastically generates multiple answers, and then ensemble them while filtering out low-quality samples to mitigate noise from inaccurate labels. Instead of utilizing the large LMs, which require substantial costs and restricted accessibility. This work focuses on smaller LMs and aims to mine their capacity to solve downstream QA tasks without fine-tuning by using external knowledge. Experiments show significant performance improvements on various QA tasks with higher robustness across diverse prompts.

**Reasons To Accept:**

1. Instead of using conventional strategies, such as Top-k or nucleus sampling, to obtain multiple self-generated answers based on a single representation, this work proposes to leverage multiple representations generated through Monte Carlo dropout, which could increase the diversity of answer sampling. This idea is simple yet effective.
2. The experiments are comprehensive and the performance gains demonstrate the effectiveness of the proposed methods.
3. The paper is clearly written and easy to follow.

**Reasons To Reject:**

1. Despite the performance gains, directly selecting the pseudo label via majority voting strategy, i.e., the highest number of occurrences, is unreliable. since the generated results of smaller LMs are less reliable compared to large LMs. There is a higher probability that all the generated answers are incorrect, and using these pseudo labels for training may hinder the model's performance. The part requires more discussion.
2. To show the effectiveness of the proposed method, it also needs to compare the performance with those large LMs (zero-shot). Even if it still cannot beat the large LMs, it would still be very encouraging if it shows a narrowing of the performance gap between smaller LMs and large LMs.
3. The author only evaluates the proposed strategy on a single backbone model. It is hard to see whether this method can be generalized to various language models or not.

**Reproducibility:**

3: Could reproduce the results with some difficulty. The settings of parameters are underspecified or subjectively determined; the training/evaluation data are not widely available.

**Reviewer Confidence:**

4: Quite sure. I tried to check the important points carefully. It's unlikely, though conceivable, that I missed something that should affect my ratings.

---

> ### Author Rebuttal · Authors · 2023-08-29
>
> >**Weakness 1)** Directly selecting the pseudo label via majority voting strategy, i.e., the highest number of occurrences, is unreliable, since the generated results of smaller LMs are less reliable compared to large LMs. The part requires more discussion.
>
>  We do not directly select the pseudo-label via majority voting. Instead, as you mentioned, we also pointed out that the generated data from smaller LMs can be unreliable (Lines 89-94), and further proposed to filter out these potentially incorrect samples having a low agreement (Lines 208-222). Also, we already provided analysis and ablation studies in Figure 3 and Table 3, showing that our proposed filtering strategy effectively abstains from training with low-quality pseudo labels, thus contributing to significantly improved performance.
>
> To be more clear, we additionally provide the effectiveness of our proposed filtering strategy for the smaller LMs with different sizes. As shown in Table B.1, all models show significant performance enhancements, and the performance gains for the smaller models are more remarkable. This highlights the importance of filtering out unreliable self-generated samples, especially for the smaller LMs. We will include them in the revision.
>
> Table B.1: Effectiveness of our filtering strategy with diverse FLAN-T5-series models on the NQ.
> |Sizes|Methods|F1|Increasement|
> |---|---|---|---|
> | Base (250M) | T-SAS w/o Filtering  | 45.00 | |
> | Base (250M) | T-SAS   | 50.22 | **+11.60%** |
> | Large (780M) | T-SAS w/o Filtering | 68.25 |  |
> | Large (780M) | T-SAS |  74.20 | **+8.72%** |
> | XL (3B) | T-SAS w/o Filtering | 73.24 |  |
> | XL (3B) | T-SAS |   75.29 | **+2.80%** |
>
> >**Weakness 2)** To show the effectiveness of the proposed method, it also needs to compare the performance with those large LMs (zero-shot).
>
> Comparing the performance with large LMs is outside of our scope, since our work targets at investigating the effectiveness of smaller LMs on self-adaption (Lines 42-47). Please note that large LMs and smaller LMs are not directly comparable, considering their discrepancy in capacity.
> However, as requested, we additionally report the performance of the large zero-shot LM (text-davinci-003) as an indicator in Table B.2. Surprisingly, our T-SAS significantly outperforms a much larger zero-shot LM, which further signifies the effectiveness of the proposed self-adaptive strategies for smaller LMs.
>
> Table B.2: Results on three QA datasets with FLAN-T5-XL based models and GPT-3 (text-davinci-003).
> |Datasets|Methods|EM|F1|
> |---|---|---|---|
> | NQ | Naïve LM | 54.53 | 66.39 |
> | NQ | GPT-3 | 55.53 | 70.57 |
> | NQ | T-SAS (Ours) |  **63.96** | **75.29** |
> | TQA | Naïve LM |  75.27 |  81.17 |
> | TQA | GPT-3 | 71.75 | 81.57 |
> | TQA | T-SAS (Ours) | **78.38** |  **84.01** |
> | SQuAD | Naïve LM |  71.66 |  83.00 |
> | SQuAD | GPT-3 | 57.16 | 75.16 |
> | SQuAD | T-SAS (Ours) | **73.84** |  **84.72** |
>
> >**Weakness 3)** The authors only evaluate the proposed strategy on a single backbone model.
>
> We already provided the performance of another one of the most widely used smaller LMs, T0-3B, in Table 5. As described in Lines 652-657, our T-SAS with T0 consistently enhances the performance across three QA datasets, which supports the applicability of T-SAS on various recent smaller LMs.

---

### Official Review · Reviewer_ruV4 · 2023-08-04

**Soundness:** 4

**Excitement:**

3: Ambivalent: It has merits (e.g., it reports state-of-the-art results, the idea is nice), but there are key weaknesses (e.g., it describes incremental work), and it can significantly benefit from another round of revision. However, I won't object to accepting it if my co-reviewers champion it.

**Missing References:**

[1] Robust Question Answering against Distribution Shifts with Test-Time Adaptation: An Empirical Study
[2] Efficient test time adapter ensembling for low-resource language varieties.
[3] PADA: Example-based prompt learning for on-thefly adaptation to unseen domains.
[4] Self-supervised test-time learning for reading comprehension.

**Paper Topic And Main Contributions:**

This paper studies test-time fine-tuning with unlabeled test data. The initial model is the one tuned to follow instructions. In this paper, it uses FLAN-series models.

The initial model will be fined tuned with unlabeled test data, which is a transductive-learning setup. In the method, the initial model has generate pseudo labels on the test data, then the pseudo labels will be used to fine-tune the model.

The paper proposes a method which samples several pseudo labels then filter the noisy labels, to make sure the training more robust.



**Questions For The Authors:**

For the experiments, have you run multiple times for each baseline?

**Reasons To Accept:**

1. The setting is practical and reasonable.
Transductive learning should be very useful in practice since it can explore the information of the test data, so it can find out more correct labels on the test data, though it is expensive to refine the model.

2. The method is sound with enough experimental results, given it is a short paper.



**Reasons To Reject:**

Though the setting is practical and reasonable, the novelty seems to be limited.
Test-time fine-tuning or adaptation has been studied for NLP tasks in the past few years, so the setting studied in this work is not new.

Related work:
[1] Robust Question Answering against Distribution Shifts with Test-Time Adaptation: An Empirical Study
[2] Efficient test time adapter ensembling for low-resource language varieties.
[3] PADA: Example-based prompt learning for on-thefly adaptation to unseen domains.
[4] Self-supervised test-time learning for reading comprehension.



**Reproducibility:**

4: Could mostly reproduce the results, but there may be some variation because of sample variance or minor variations in their interpretation of the protocol or method.

**Reviewer Confidence:**

5: Positive that my evaluation is correct. I read the paper very carefully and I am very familiar with related work.

---

> ### Author Rebuttal · Authors · 2023-08-29
>
> >**Weakness 1-1)** The novelty seems to be limited.
>
> We strongly believe that our work is novel and that its contributions are substantial, which we summarize as follows:
>
> * First of all, we introduce **a novel and realistic problem** by exploring the self-adaptive capabilities of the **recently proposed smaller LMs** (See Lines 37-47), specifically in a realistic test-time setting where the labeled data is unavailable.
> * Then, we address this problem with two novel and effective strategies (Section 3.2): self-ensemble strategy with stochastic generation via majority voting (Lines 180-207) and filtering strategy which removes samples with low-quality according to their agreement (Lines 208-222).
> * We experimentally showed that our proposed T-SAS significantly improves the performance and that each of the proposed self-ensemble and filtering strategies is effective (Table 3). Furthermore, our T-SAS is substantially robust to diverse prompts (Lines 281-292 and Lines 676-689), which effectively addresses an undesirable but realistic prompt-sensitive problem in recent LMs.
>
>
> >**Weakness 1-2)** Some related work exists [A1-A4].
>
> We thank you for your suggestion on the related work. However, the suggested work and ours are **largely different**, since the previous studies focus on solving classification problems under the **extractive setting** mainly using BERT series models. In contrast, ours focuses on the **generative setting**. Please note that as the objectives of extractive and generative settings are different, the test-time adaption strategies are fundamentally different. To be specific, in an extractive setting, self-adaptation is based on probabilities, while in a generative setting, self-adaptation is done using generated text. In situations where filtering is further applied, in the extractive setting, filtering is based on probabilities, whereas in the generative setting, it is done using the generated text (We discovered that it is less effective to filter out samples based on a single generation probability in Table 3). This makes it difficult to directly adopt the previous test-time adaptive strategies used for the extractive models to the generative setting. Therefore, while several previous work has investigated test-time adaption approaches to solving classification tasks under the extractive setting, it has been questionable and largely underexplored how we can self-adopt the small LMs over the generative setting.
>
> Also, please note that the difference in extractive and generative settings makes it difficult to directly compare our generative T-SAS against the extractive models through experiments.
>
> We concretely further discuss the differences between each of the suggested papers and ours as follows:
>
> [A1] Robust Question Answering against Distribution Shifts with Test-Time Adaptation: An Empirical Study
> * This work targets an **Extractive** task (QA), which is largely different from our generative setting, as aforementioned above.
> * Also, since this work aims to correctly classify the tokens in an extractive setting, the filtering approach in this work is based on a single token probability. Please note that such a filtering approach applied to our generative setting is already presented as a baseline in Table 3 (w/o Stochastic), which indicates the effectiveness of our stochastic filtering strategy.
>
> Although conducting a direct apples-to-apples comparison with this work is infeasible due to the distinct extractive/generative natures of two different settings, we have further compared our T-SAS against the suggested model, OIL, along with other baseline models PL and Tent in this work. Specifically, we tried our best to make a fair assessment by 1) comparing models with similar parameter sizes (FLAN-T5-base (250M) vs XLM-RoBERTa-base (270M)) and 2) finetuning all the models with the SQuAD dataset before the test-time adaption. As shown in Table A.1, our T-SAS significantly outperforms the extractive baseline models.
>
> Table A.1: Comparison between the extractive models from [A1] and our T-SAS on the NQ.
> |Methods|EM|F1|
> |---|---|---|
> |Naïve LM|31.72|40.84|
> |Tent|43.14|56.11|
> |Tent+xTune|44.98|59.67|
> |PL|43.53|56.32|
> |PL+xTune|44.89|59.53|
> |OIL|43.00|56.44|
> |OIL+xTune|45.30|60.16|
> |T-SAS (Ours)|**58.27**|**69.29**|
>
> [A2] Efficient test time adapter ensembling for low-resource language varieties.
> * This work targets at an **Extractive** task (cross-lingual NER and POS tagging), which is largely different from our generative QA setting, as aforementioned above.
> * This work aims at ensembling multiple models for diverse languages to robustly classify tokens in an unseen language. In contrast, instead of ensembling multiple different models, our self-ensemble strategy considers self-generated outputs from a single model.
>
> [A3] PADA: Example-based prompt learning for on-the-fly adaptation to unseen domains.
> * This work does not target at a **Test-Time Adaption** task, but rather aims at generating an appropriate prompt for a test sample, without further training the model during the test-time.
> * This work targets at an **Extractive** task (text classification and sequence tagging), which is largely different from our generative setting, as aforementioned above.
>
> [A4] Self-supervised test-time learning for reading comprehension.
> * This work targets an **Extractive** task (QA), which is largely different from our generative setting, as aforementioned above.
> * The direction of the test-time adaption is different and rather orthogonal to ours. Specifically, our test-time adaption focuses on effectively addressing the given test-time input questions by generating synthetic answers. However, the previous work aims at generating synthetic question-answer pairs based solely on the provided test-time context, without taking into account the actual test-time input questions. As a result, direct adaptation to the test-time input questions may not be feasible in this setting, given that the adapted model does not have visibility into the actual test-time input questions.
> * In addition, this work does not consider the quality of the synthetically generated data. In contrast, we propose self-ensemble and filtering strategies to ensure training models with high-quality data.
>
> >**Question 1)** For the experiments, have you run multiple times for each baseline?
>
> Yes, we have run each baseline multiple times, by using different prompts, as shown in Figure 2. Specifically, we run each model with 11 different prompts and Figure 2 illustrates the average F1 scores with corresponding standard deviation. Figure 2 indicates that our model not only consistently outperforms other baselines, but also is highly robust to different prompts (Lines 286-292).
> Additionally, we further report the average and standard deviation on the NQ over three different runs in Table A.2, where we use the same prompt with different seeds. Here, we observe that our T-SAS shows clear performance improvements over the baselines and is statistically significant at p < 0.01.
>
> Table A.2: Three runs on the NQ using the FLAN-T5-XL models.
> |Methods|EM|F1|
> |---|---|---|
> |Self-Adaptive w/ Greedy | 59.31 &pm; 0.41 | 70.86 &pm; 0.35 |
> |Self-Adaptive w/ Soft | 57.85 &pm; 0.35 | 70.56 &pm; 0.44|
> |Self-Adaptive w/ LMSI | 62.11 &pm; 0.68 | 73.74 &pm; 0.40|
> |T-SAS (Ours) | **63.91 &pm; 0.09**|**75.20 &pm; 0.12**|

---

### Meta-Review · Area_Chair_9rWr · 2023-09-09

**Recommendation:** 4

**Metareview:**

The paper discusses a self-learning strategy for fine-tuning relatively small LM for Question Answering tasks using only unlabelled data.
The proposed solution is relatively simple but sound and intuitive and the results observed on multiple datasets are remarkable.

There is enough substance for a short paper.
Related works should be better discussed.

---

### Decision · Program_Chairs · 2023-10-07

**Decision:**

Accept-Findings

**Comment:**

The paper discusses a self-learning strategy for fine-tuning relatively small LM for Question Answering tasks using only unlabelled data.
The proposed solution is relatively simple but sound and intuitive and the results observed on multiple datasets are remarkable.

There is enough substance for a short paper.
Related works should be better discussed.